# Transient Elastography as the First-Line Assessment of Liver Fibrosis and Its Correlation with Serum Markers

**DOI:** 10.3390/medicina59040752

**Published:** 2023-04-12

**Authors:** Nikola Uzlova, Katerina Mnozil Stridova, Dusan Merta, Ivan Rychlik, Sona Frankova

**Affiliations:** 1Department of Internal Medicine, University Hospital Královské Vinohrady, 100 34 Prague, Czech Republic; 2Third Faculty of Medicine, Charles University, 100 00 Prague, Czech Republic; 3Anaesthesiology and Resuscitation Department, Institute for Clinical and Experimental Medicine, 140 21 Prague, Czech Republic; 4First Faculty of Medicine, Charles University, 120 00 Prague, Czech Republic; 5Department of Hepatogastroenterology, Institute for Clinical and Experimental Medicine, 140 21 Prague, Czech Republic

**Keywords:** vibration-controlled transient elastography, ELF test, APRI, FIB-4, liver stiffness, advanced fibrosis

## Abstract

*Background and objectives*: Recently, rapid progress has been made in the development of noninvasive methods for liver fibrosis assessment. The study aimed to assess the correlation between LSM and serum fibrosis markers to identify patients with advanced liver fibrosis in daily clinical practice. *Methods*: Between 2017 and 2019, 89 patients with chronic liver disease of various etiology, 58 males and 31 females, were enrolled in the study and underwent ultrasound examination, vibration-controlled transient elastography (VCTE), AST to Platelet Ratio Index (APRI score), Fibrosis-4 (FIB-4) score, and enhanced liver fibrosis (ELF) test. *Results*: The diagnoses were as follows: NAFLD (30.3%), HCV (24.3%), HBV (13.1%), ALD (10.1%), other (7.8%). Their median age was 49 (21–79), and their median BMI was 27.5 (18.4–39.5). The median liver stiffness measurement (LSM) was 6.7 kPa (2.9–54.2 kPa), the median of the ELF test was 9.0 (7.3–12.6), and the median APRI was 0.40 (0.13–3.13). Advanced fibrosis assessed by LSM was present in 18/89 (20.2%) patients. The LSM values correlated with the ELF test results (r^2^ = 0.31, *p* < 0.0001), with the APRI score (r^2^ = 0.23, *p* < 0.0001), the age of the patients (r^2^ = 0.14, *p* < 0.001), and with the FIB-4 values (r^2^ = 0.58, *p* < 0.0001). The ELF test values correlated with the APRI score (r^2^ = 0.14, *p* = 0.001), the age (r^2^ = 0.38, *p* < 0.0001), and the FIB-4 (r^2^ = 0.34, *p* < 0.0001). By determining the confidence intervals of the linear model, we proved that patients younger than 38.1 years have a 95% probability of absence of advanced liver fibrosis when assessed by VCTE. *Conclusions*: We identified APRI and FIB-4 as simple tools for screening liver disease in primary care in an unselected population of patients. The results also showed that individuals younger than 38.1 years had a negligible risk of advanced liver fibrosis.

## 1. Introduction

Liver fibrosis is an inevitable consequence of chronic liver diseases and a significant cause of morbidity and mortality in the industrial world [1]. In 2017, chronic liver disease (CLD) affected a staggering 1.5 billion people worldwide, with the most common causes being non-alcoholic fatty liver disease (NAFLD) (60%), hepatitis B virus (HBV) (29%), hepatitis C virus (HCV) (9%), and alcoholic liver disease (ALD) (2%) [2]. Despite the high prevalence of CLD, data on the prevalence of cirrhosis in resource-limited settings, particularly in regions outside of Europe, are scarce [3]. However, in European countries, the median cirrhosis prevalence was 833 cases per 100,000 individuals, with a range of 447 to 1100 cases per 100,000 individuals.

As obesity rates continue to rise worldwide, the prevalence of NAFLD and its complications are increasing as well. The stage of fibrosis, which refers to the extension of liver scarring, appears to be a significant factor associated with mortality in patients with NAFLD [4]. Therefore, it is crucial to identify patients who have advanced fibrosis to ensure they receive the necessary medical attention and treatment. Overall, the high prevalence of CLD and its various causes highlight the importance of continuous research and public health efforts to prevent and treat this condition. Identifying patients with advanced fibrosis is just one example of how we can improve patient outcomes and reduce the burden of CLD on individuals and healthcare systems worldwide.

The degree of liver fibrosis is an important prognostic factor. Early assessment of liver fibrosis plays a crucial role in patients management [5]. Liver biopsy is an essential tool in diagnosing liver disease and determining the presence of liver fibrosis. However, it requires obtaining a sufficient sample of liver tissue, and its evaluation requires expertise to provide reliable information. Despite providing detailed information, these limitations highlight the need for alternative noninvasive diagnostic methods [6]. While the liver biopsy is a reliable diagnostic tool, it is also an invasive procedure and carries a risk of serious complications, albeit rare. Due to the associated financial burden and risk, it is desirable to develop noninvasive, easily repeatable, and cost-effective alternatives for assessing the progression of liver fibrosis. This approach would enhance patient safety and comfort, enable serial monitoring of the disease progression, and facilitate the implementation of appropriate treatment strategies. The process of liver fibrogenesis is dynamic, and with time, fibrosis may also regress [7]. By utilizing noninvasive methods, it becomes possible to monitor the development or the fibrosis regression over time, thereby enabling the determination of the patient’s prognosis. This approach facilitates the assessment of the disease stage and allows clinicians to adjust management strategies accordingly.

In the last two decades, rapid progress has been made in the development of noninvasive methods for liver fibrosis assessment. In the past, liver biopsy represented the gold standard for the staging of liver diseases; as an invasive method, it should be used nowadays only to confirm the etiology of liver disease.

In hepatology, vibration-controlled transient elastography (VCTE) represents the most used and validated method, based on the principle of mechanical energetic impulse. The Fibroscan^®^ device (Echosens, Paris, France) works on the principle of VCTE and measures liver stiffness (liver stiffness measurement, LSM). The method was introduced into clinical practice in 2003 and has been validated by many studies across the entire spectrum of liver disease etiologies. After 20 years of use, it is a routine method in daily clinical hepatology practice [1]. It has become a valid alternative to liver biopsy in staging liver disease, including liver cirrhosis. Yet, despite its reliability, the method has some limitations. The examination of obese patients can be overcome using XL probes with higher penetrance. The results may also be influenced by the patient’s habitus, the operator’s experience, a high grade of inflammation, cholestasis, and venous congestion after a meal, or, when the patient suffers from portal vein thrombosis [8].

Together with the progress of imaging-based noninvasive methods for liver fibrosis assessment, many specific and non-specific serum markers of fibrogenesis have been investigated. Their use in mathematical models and scores has become an integral component of noninvasive screening tools (noninvasive tests, NITs) in routine clinical practice. Each of these mathematical models correlates with a specific etiology of liver disease: the basic, cheap, and most widely used tests are AST to Platelet Ratio Index (APRI) and the Fibrosis-4 (FIB-4) score. However, the results may fall into the area of uncertainty in 30–50% of patients [9]. One of the comprehensive NITs is the enhanced liver fibrosis (ELF) test, which is based on testing markers of acute phase reactants, such as hyaluronic acid (HA), type III procollagen peptide (PIIINP) and tissue inhibitor of metalloproteinase-1 (TIMP1). As validated by a meta-analysis, this test bears a high sensitivity. Still, its specificity is limited to patients with non-alcoholic fatty liver disease (NAFLD/NASH) in determining severe fibrosis [10].

Liver cirrhosis or severe fibrosis are considered to be advanced liver disease, i.e., the F3 stage according to the Metavir score, or liver stiffness of more than 10 kPa on transient elastography assessment, or a value of more than 9.8 on the ELF test [11].

To date, there has been no study comparing VCTE, the worldwide standard method for investigating the stage of liver fibrosis, with noninvasive laboratory tests commonly used to assess the risk of fibrosis and serum markers of fibrogenesis, in the Czech Republic. This is perhaps due to the fact that VCTE (Fibroscan^®^ device) is available only in a few centers in the country. In the clinical practice, hepatologists in the Czech Republic often encounter patients who come to be examined at advanced stages of chronic liver disease or, on the contrary, too early. The lack of a standardized and widely available method for assessing liver fibrosis contributes to this issue.

The study aimed to assess the correlation between LSM and serum fibrosis markers, in order to identify patients with advanced liver fibrosis in the daily clinical practice at the first contact with patients with chronic liver disease. Conducting a study comparing transient elastography with noninvasive laboratory tests in the Czech Republic would provide valuable insight into the effectiveness of these methods of assessing the risk of fibrosis and serum markers of fibrogenesis. The implementation of a standardized and widely available method of assessing liver fibrosis would allow earlier detection and intervention, ultimately improving patient outcomes, and reducing the burden of chronic liver disease on individuals and healthcare systems in the Czech Republic.

## 2. Materials and Methods

The prospective study was conducted at the outpatient department of the Department of Internal Medicine of the University Hospital Královské Vinohrady, Prague, Czech Republic, between 2017 and 2019. The study was approved by the Ethics Committee and was carried out in compliance with the Helsinki Declaration. All study participants gave written consent to the storage of blood samples and agreed to use blood samples for future research.

Eighty-nine consecutive patients referred for their first fibrosis assessment by general practitioners, hepatologists, and infectious disease specialists from Prague and Central Bohemia Region were enrolled in the study.

The study population consisted of individuals with chronic liver disease, as assessed by the referring physician, over the age of 18, who had undergone a valid VCTE examination and had provided their consent to be included in the study. The participants were included consecutively, without prior knowledge of the cause of their liver disease, ensuring that the study population was unselected. The exclusion criteria for the study were: clinical symptoms of acute hepatitis, biliary tree obstruction, heart failure with congestive liver disease, and the inability to perform a VCTE examination; however, none of the participants had to be excluded for these reasons.

All the patients signed an informed consent to participate in the study before any study procedure. The cohort included patients with alcoholic liver disease (ALD), chronic hepatitis C (HCV), chronic hepatitis B (HBV), HCV/HIV coinfection, non-alcoholic fatty liver disease (NAFLD), primary sclerosing cholangitis (PSC), Wilson’s disease, or so far undetermined liver disease. All the patients underwent ultrasound examination, and on the same day, they provided their medical history data, underwent VCTE (FibroScan 502 and Compact 530, Echosens, Paris, France), and blood tests for the assessment of the serum fibrosis markers (APRI score, FIB-4 score, and ELF test). The controlled attenuation parameter (CAP) to assess the stage of liver steatosis was evaluated in 48 patients in whom LSM was performed.

### 2.1. Vibration-Controlled Transient Elastography

The LSM was performed using VCTE (Fibroscan^®^, Echosens, Paris, France). One experienced examiner, unaware of the serum fibrosis test results, completed all the examinations [12] after three hours of the patient’s fasting [13]. The M or XL probes were used appropriately according to the body mass index of the examined subject. The LSM value is given in kPa, with a range of 2.5–75 kPa. The quality of the examination was defined, according to the instruction of the manufacturer, as ten valid measurements with IQR/median ≤ 30% [14,15]. The LSM value of 10 kPa was considered to be the cut-off value for advanced liver fibrosis [16].

### 2.2. Laboratory Assessments

The following laboratory parameters were assessed after at least three hours of fasting period, on the same day of the transient elastography examination: total bilirubin, aspartate aminotransferase (AST), alanine aminotransferase (ALT), alkaline phosphatase (ALP), γ-glutamyltransferase (GGT), albumin, and platelet count.

The pre-analytic and analytic phases of the laboratory samples analysis were conducted in accordance with the standards of the Central Laboratory of the University Hospital Královské Vinohrady (https://www.fnkv.cz/lab/lp_uld/_start.htm, accessed on 4 July 2022).

### 2.3. ELF Test

ELF test is a noninvasive blood test measuring three direct markers of liver fibrosis: hyaluronic acid (HA), procollagen type III N-terminal peptide (PIIINP), and tissue inhibitor of metalloproteinase I (TIMP1). The blood was drawn at the same time as the blood for the other laboratory tests, and the serum was stored at −80 °C until the analysis was performed. The testing was performed using an ADVIA Centaur XP (Siemens Healthcare Diagnostics Inc., Westchester, NY, USA). The results were calculated using the algorithm provided by the manufacturer, ELF score = 2.278 + 0.851 ln(HA) + 0.751 ln(CPIIINP) + 0.394 ln(CTIMP1) [17]. The value of the ELF test 10.51 was considered to be the cut-off value for advanced fibrosis.

### 2.4. APRI and FIB-4

APRI and FIB-4 represent the basic biochemical scores.

APRI was calculated according to the following formula [18]:$$\frac{\;\mathrm{AST}\;\mathrm{IU}/L/\mathrm{AST}\;\mathrm{ULN}}{\mathrm{platelets}\;10\times 10^{9}/L\;}\times \;100$$

Advanced fibrosis is considered improbable when the values are <0.5 and highly probable when the values reach >1.5 [1].

FIB-4 was calculated according to the following formula [19]:$$\frac{\mathrm{age}\;\left(\mathrm{years}\right)\;\times \;\mathrm{AST}\;\left(\mathrm{IU}/L\right)}{\mathrm{platelets}\;10\times 10^{9}/L\;\times \;\sqrt{\mathrm{ALT}}}$$

Advanced fibrosis is considered improbable when the values are <1.45 and highly probable when the values reach >3.25 [1].

### 2.5. Statistical Analysis

Quantitative data are presented as medians and ranges, and qualitative data are reported as percentages (%). The Shapiro–Wilk test was used to evaluate the normal distribution of the data. Medians were compared using the *t*-test, Mann-Whitney test, or Kruskal-Wallis test, as appropriate. Linear regression was applied to assess correlations of the studied values, and the demographic data and the factors predicting the presence of advanced fibrosis were examined using multivariate logistic regression analysis.

A *p*-value of <0.05 indicated statistical significance for all calculations. The statistical analysis was performed using the GraphPad Prism version 9.4.0 for Mac, GraphPad Software, San Diego, CA, USA, www.graphpad.com, accessed on 4 July 2022, and R programming language version 4.1.3 (www.r-project.org, accessed on 4 July 2022).

## 3. Results

Eighty-nine patients, 58 males and 31 females, were enrolled in the study. The baseline demographic data are presented in Table 1. The diagnoses were as follows: NAFLD/NASH (30, 33.7%), HCV (24, 27.0%), HBV (13, 14.6%), ALD (10, 11.2%), HCV/HIV (5, 5.6%), PSC (1, 1.1%), Wilson’s disease (1, 1.1%), and unknown (5, 5.7%). The median age of the patients was 49 years (range 21–79), with a median body mass index (BMI) of 27.5 (18.4–39.5).

The median of the LSM was 6.7 kPa (2.9–54.2 kPa), the median value of the ELF test was 9.0 (range 7.3–12.6), and the median APRI was 0.40 (range 0.13–3.13). Advanced fibrosis assessed by LSM was present in 18/89 (20.2%) patients.

### 3.1. LSM and NITs Correlations

The LSM values correlated with the ELF test results (r^2^ = 0.31, *p* < 0.0001); the values were in correlation for both males (r^2^ = 0.31) and females (r^2^ = 0.36). The LSM values of the whole cohort correlated with the APRI score (r^2^ = 0.23, *p* < 0.0001); the values were in correlation for males (r^2^ = 0.24, *p* < 0.001), but not for females (r^2^ = 0.16, N.S.).

The LSM correlated with the age of the patients (r^2^ = 0.14, *p* < 0.001); the values were in correlation for males (r^2^ = 0.12, *p* < 0.01) and for females (r^2^ = 0.16, *p* = 0.02).

The LSM values correlated with the FIB-4 values (r^2^ = 0.58, *p* < 0.0001), the values were in correlation for males (r^2^ = 0.6, *p* < 0.0001) and for females (r^2^ = 0.47, *p* = 0.002) (Figure 1).

The ELF test values correlated with the APRI score (r^2^ = 0.14, *p* = 0.001). The values were in correlation for males (r^2^ = 0.12, *p* = 0.01) as well as for females (r^2^ = 0.18, *p* = 0.04). The ELF test values correlated with age (r^2^ = 0.38, *p* < 0.0001). The values were in correlation for males (r^2^ = 0.37, *p* < 0.0001) and for females (r^2^ = 0.38, *p* = 0.0001). The ELF test values correlated with the FIB-4 (r^2^ = 0.34, *p* < 0.0001); the values were in correlation for males (r^2^ = 0.32, *p* < 0.0001) and for females (r^2^ = 0.50, *p* = 0.006) (Figure 2).

### 3.2. LSM and NITs Correlations According to Liver Disease Etiology

The values of noninvasive serum tests and the results of the LSM did not differ significantly between the groups of patients according to the liver disease etiology (Table 2).

In patients with ALD, the LSM values correlated with the ELF test results (r^2^ = 0.66, *p* = 0.04) and the APRI (r^2^ = 0.74, *p* = 0.02), but did not correlate with age owing to the small cohort size (10 patients).

We found a correlation in all analyzed variables in the group of patients with viral hepatitis: LSM and age (r^2^ = 0.56, *p* < 0.0001), LSM and ELF (r^2^ = 0.60, *p* < 0.0001), LSM and APRI (r^2^ = 0.42, *p* = 0.02), and LSM and FIB-4 (r^2^ = 0.62, *p* = 0.0003).

The values correlated in the NAFLD/NASH patients for LSM and age (r^2^ = 0.46, *p* = 0.01), LSM and ELF (r^2^ = 0.58, *p* = 0.0008), but the correlation was not statistically significant between LSM and APRI, and LSM and FIB-4.

The group of other diagnoses, consisting of mere seven patients, was not evaluated owing to the small sample size.

### 3.3. Factors Predicting Advanced Liver Fibrosis

By the determination of the confidence intervals of the linear model, we proved that patients younger than 38.1 years have a 95% probability of absence of advanced liver fibrosis (>10 kPa) when liver fibrosis is assessed by VCTE (Figure 3).

The age cut-offs were also calculated separately for the two largest sub-groups of patients in the study; in the groups of patients with viral hepatitis and NAFLD/NASH, the cut-off points for age predicting the high probability of absence of advanced fibrosis were 39.0 years and 35 years, respectively (Appendix A).

Furthermore, the factors predicting the presence of advanced fibrosis in the entire cohort of patients were analyzed. Only age was the factor predicting the presence of advanced fibrosis at the first hepatologic examination in our cohort of patients (OR 1.05 for one year increment, 95% CI 1.0–1.1). Neither the female sex (OR 0.83, 95% CI 0.24–2.63), nor BMI > 30 (0.83, 95% CI 0.58–6), or diagnosis of NAFLD/NASH (OR 1.37, 95% CI 0.39–4.63) predicted the presence of advanced liver fibrosis (Figure 4).

## 4. Discussion

Our cohort of patients was unselected and referred to their first hepatological examination by primary care physicians, hepatologists, and infectious disease specialists. This population group reflects the distribution of liver disease etiology in the general population in western countries [20].

At their first hepatological examination, we identified advanced liver fibrosis in 18/89 (20.2%) patients [21]. In a French study, screening the stage of liver disease in a general population older than 45 years, advanced fibrosis, or cirrhosis (F3 and F4) were found in 98 (8.2%) subjects out of 1190 subjects screened by VCTE. NAFLD was the most common cause of underlying liver disease (52 patients). The LSM turned out to be a valuable and specific tool to screen cirrhosis in the general population and to detect undiagnosed chronic liver disease in these so far healthy individuals.

In our cohort, an LSM value lower than 10 kPa was considered to exclude the presence of advanced liver disease according to the Baveno VI Consensus recommendations. However, a multicenter study published in 2021 resulted in the redefinition of the dual cut-offs for the LSM, with values below <8 kPa (or <7 kPa for viral hepatitis) and above 12 kPa showing better diagnostic accuracy in compensated advanced chronic liver disease (cACLD), compared to the previous cut-offs of <10 kPa and >15 kPa recommended by the Baveno VI Consensus [16]. The findings indicate that a considerable proportion of younger patients could be classified as having advanced liver disease, emphasizing the importance of early diagnosis to prevent disease progression and the development of complications such as liver cirrhosis decompensation and hepatocellular carcinoma. Therefore, prompt detection and management of liver disease in this age group are critical to improve the prognosis and reduce the associated morbidity and mortality.

In the Czech Republic, VCTE by Fibroscan^®^ is not widely available to be used as a screening tool in the general population. It is available only in large volume hepatological centers focused on the treatment of viral hepatitis. Therefore, the noninvasive serum markers could represent a reliable diagnostic alternative in the first-line assessment.

In our unselected cohort of patients with various liver diseases, the serum markers of fibrosis showed a good correlation with the examination by VCTE. Based on our results, we identified APRI and FIB-4 as simple tools for screening liver disease in primary care, independent of the etiology of liver disease. These tests are based on routinely performed and financially affordable biochemical and blood parameters, routinely available as part of regular medical and preventive examinations. In a small study conducted in 2019, the APRI/FIB-4 combination performed well in predicting significant liver fibrosis, while the FIB-4 alone did well in predicting cirrhosis [22]. These noninvasive biochemical markers could be used as a screening tool instead of the LSM.

The individual methods are usually validated for particular etiologies of liver disease based on the data published in previous studies [5]. However, the study performed by Innes et al. analyzed the performance of routine risk scores to predict long-term cirrhosis complications in the general population. The APRI score exhibited the highest discriminative ability, followed by the FIB-4 score [23].

Our results also showed that individuals older than 38.1 years, independent of sex, are at a higher risk of liver disease, especially in the group of patients with NAFLD/NASH and metabolic syndrome.

This finding is particularly notable, as age is not typically considered a significant risk factor for liver fibrosis. Previous studies have identified a spectrum of risk factors for liver fibrosis, including obesity, presence of diabetes, and regular alcohol consumption. However, the association between age and liver fibrosis is less well-established. Considering the pathophysiological mechanisms of liver fibrosis development, which is determined by, among other factors, the length of time of the presence of liver disease and the amplifying factors [24], we can conclude that the occurrence of liver disease shifts to younger age.

In 2020, a UK population-based study was published with the aim of identifying the presence of steatosis and fibrosis in individuals aged from 18 to 35 years using VCTE. The study involved 4021 participants with a mean age of 24 years. One in five young people had steatosis, and one in forty had significant fibrosis around the age of 24 years [25].

The aim of a U.S. study was to assess the prevalence of alcohol-associated fatty liver disease (ALD) and non-alcoholic fatty liver disease (NAFLD) in a representative U.S. cohort utilizing VCTE to measure hepatic steatosis and suspected fibrosis in adolescents and young adults (age between 15 and 39 years). In the group of 1319 participants, ALD was present in 56.59% of examined individuals. In those with suspected ALD, significant fibrosis (F ≥ F2) was present in 12.3%, and severe fibrosis (F ≥ F3) was present in 6.31% of the enrolled patients. Similarly, in subjects without excessive alcohol consumption, suspected NAFLD was present in 40.04%. In those with suspected NAFLD, significant fibrosis (F ≥ F2) was present in 31.07%, and suspected advanced fibrosis (F ≥ F3) was present in 20.15% [26]. Based on the existing strong evidence that alcohol-related liver disease and NAFLD shift to younger age, we can presume an increasing incidence of advanced liver disease and related complications also in younger individuals. These patients are, therefore, at a higher risk of progression of liver disease and its complications, i.e., HCC [27,28].

The early identification of individuals with advanced liver fibrosis is crucial to assess the patient’s prognosis and to establish the appropriate therapeutical approach, including enrolment into the HCC surveillance programs. Based on published data, it is feasible to predict the risk of advanced liver disease without the need to adopt costly, proprietary, or invasive methods. The scores can be easily calculated from the data collected for other purposes [23].

The best approach is to diagnose chronic liver disease before it becomes symptomatic. In the Czech Republic, the system of compulsory health insurance guarantees preventive medical assessments at regular biennial intervals from the age of 18 years, which can identify individuals at risk of liver disease. Furthermore, this approach could refer these selected patients to a specialized hepatological examination.

NAFLD/NASH has become the most common cause of liver disease, and according to the meta-analyses published in 2019 and 2022, its prevalence reaches 29.8–32.4% of the global population. It is continuing to increase at an alarming rate [20,29]. One in five individuals with NAFLD present with NASH [30] with progressive fibrosis representing the risk of chronic liver failure and HCC [31,32]. The prevalence is higher in patients with comorbidities, such as type 2 diabetes (55.5%). Also, in our cohort of patients, NAFLD represented the most prevalent etiology of liver disease. Despite being a highly prevalent liver disease that can lead to severe health, economic, and social consequences, NAFLD was found to be receiving far too little attention in national health agendas globally [33].

The limitation of our study is the relatively small sample size of the study population and the absence of knowledge of more demographic characteristics of the individual included in the study, such as the presence of diabetes or drinking habits. These factors could also be included in the analysis of predictors of advanced fibrosis. Nevertheless, the study provides important insights into the prevalence of liver fibrosis in the study population in the Czech Republic. Future studies with larger sample sizes may help to validate these pilot findings further.

## 5. Conclusions

We identified APRI and FIB-4 as simple tools for screening liver disease in primary care in an unselected population of patients, correlating with LSM results. The study provides evidence that there is correlation between LSM and serum fibrosis markers, even in an unselected cohort of patients, independently of the etiology of liver disease. The results also showed that individuals younger than 38.1 years had a negligible risk of advanced liver fibrosis.

The serum fibrosis markers could serve as a screening tool in an unselected patient population in primary care and help specialists who take care of patients at risk of liver diseases, such as diabetologists, cardiologists, and bariatricians.

## Figures and Tables

**Figure 1 medicina-59-00752-f001:**
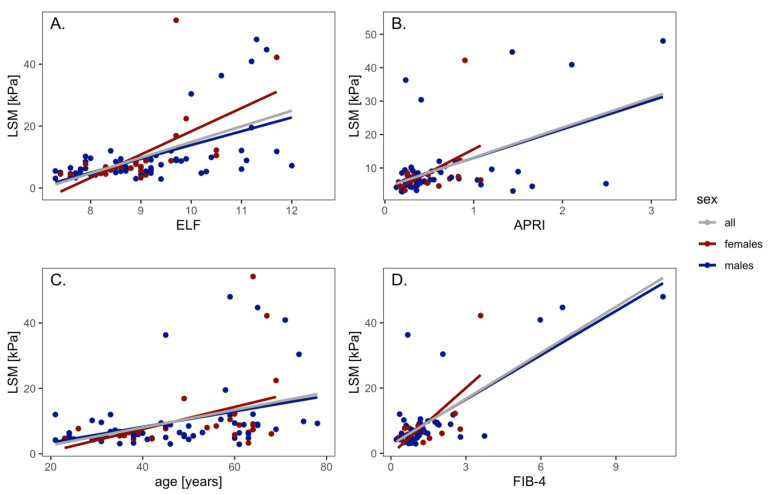
Correlation of LSM with ELF test (**A**), APRI (**B**), age (**C**) and FIB-4 (**D**).

**Figure 2 medicina-59-00752-f002:**
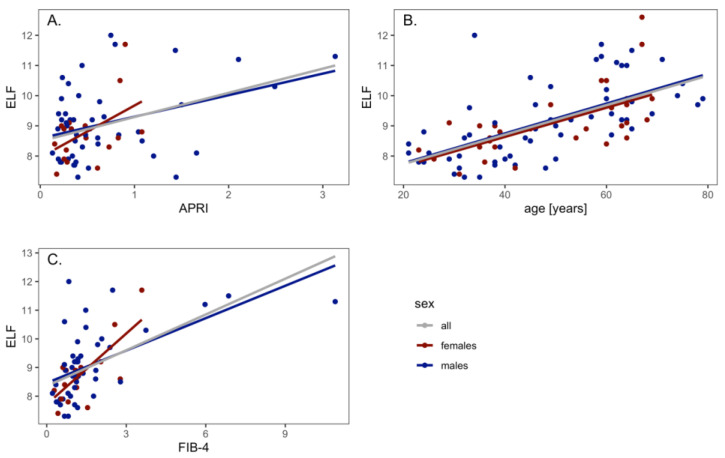
Correlation of ELF test with APRI (**A**), age (**B**) and FIB-4 (**C**).

**Figure 3 medicina-59-00752-f003:**
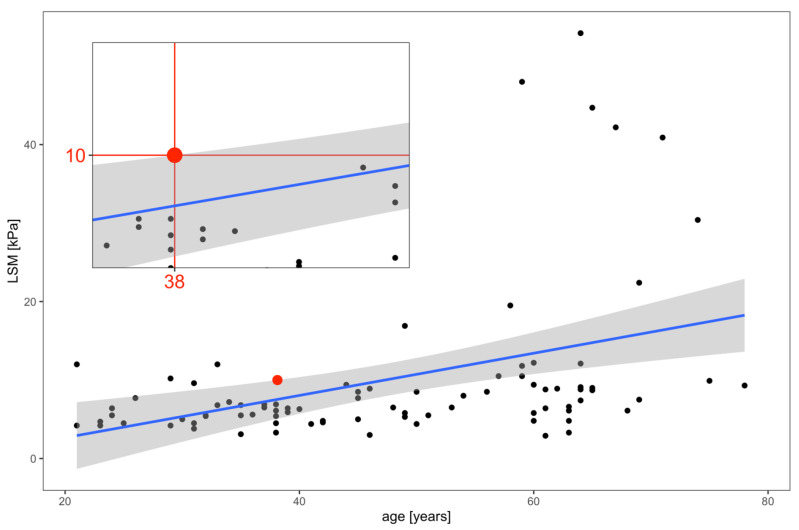
Correlation of LSM and age. Advanced liver fibrosis is unlikely in individuals younger than 38.1 years. The blue line represents the linear regression model of the correlation between LSM and age. The black points represent the individual patients. The red point represents the cut-off value with the 95% certainty that individuals younger than 38.1 years do not have LSM > 10 kPa. The confidence intervals are in grey. LSM, liver stiffness measurement.

**Figure 4 medicina-59-00752-f004:**
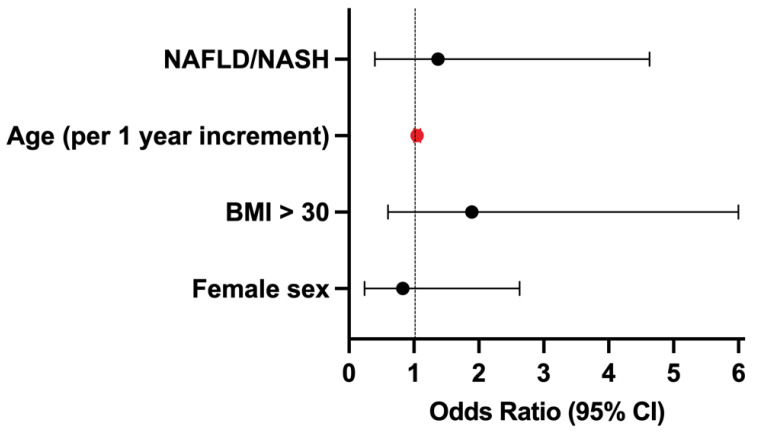
Predictors of advanced fibrosis at the first examination. CI, confidence interval. The significant predictor is depicted in red.

**Table 1 medicina-59-00752-t001:** Baseline characteristics of the cohort.

Variables		Females (*n* = 31)	Males (*n* = 58)	All (*n* = 89)
Age, years (median, range)		54 (23–69)	47 (2179)	49 (21–79)
Weight, kg (median, range)		76.2 (4–125)	86.5 (58–138)	84 (49–138)
Height, cm (median, range)		165 (150–178)	178 (160–197)	173 (150–197)
BMI, kg/m^2^ (median, range)		26.8 (18.4–39.5)	27.7 (20.5–39)	27.6 (18.4–39.5)
Waist circumference, cm (median, range)		87 (66–123)	99 (78–144)	96.5 (66–144)
	Liver disease etiology			
	NAFLD/NASH	11 (35.5%)	19 (32.8%)	30 (33.7%)
	Viral hepatitis (HBV, HCV, HCV/HIV)	16 (51.6%)	26 (44.8%)	42 (47.2%)
	ALD	2 (6.45%)	8 (13.8%)	10 (11.2%)
	Other and unknown	2 (6.45%)	5 (8.6%)	7 (7.9%)
Laboratory Data				
ALT, IU/L, median, range		51.5 (16.3–161.5)	45.2 (12.7–516.9)	46.9 (12.6–516.9)
AST, IU/L, median, range		40.7 (16.3–87.4)	34.9 (15.7–236.1)	36.8 (15.7–236.1)
GGT, IU/L, median, range		51.3 (16.2–160.8)	45.0 (12.6–514.7)	46.8 (12.6–514.7)
ALP, IU/L, median, range		71.4 (32.4–159.6)	75.0 (39.0–330.5)	72.9 (32.4–330.6)
Total bilirubin, mg/dL, median, range		0.45 (0.2–1.0)	0.39 (0.3–1.9)	0.51 (0.2–1.8)
Platelets [×10^9^/L], median, range		224 (122–398)	218 (66–483)	224 (66–483)
Noninvasive Serum Tests				
	APRI score	0.35 (0.15–1.08)	0.48 (0.13–3.13)	0.40 (0.13–3.13)
	FIB-4 score	1.1 (0.28–3.54)	1.12 (0.21–10.75)	1.11 (0.21–10.75)
	ELF score	9 (7.4–12.6)	8.95 (7.3–12)	9 (7.3–12.6)
VCTE Assessment (Fibroscan^®^)				
LSM [kPa]		6.9 (3.3–8.5)	5.5 (2.9–48)	6.7 (2.9–54.2)

ALD, alcoholic liver disease; ALT, alanine aminotransferase; ALP, alkaline phosphatase; AST, aspartate aminotransferase; APRI, AST to Platelet Ratio Index; BMI, body mass index; GGT, gamma-glutamyl transpeptidase; HBV, hepatitis B virus infection; HCV, hepatitis C virus infection; HCV/HIV, hepatitis C and human immunodeficiency virus coinfection); LSM, liver stiffness measurement; NAFLD/NASH, non-alcoholic fatty liver disease/non-alcoholic steatohepatitis; VCTE, vibration-controlled transient elastography. The reference upper limits of normal for liver function tests are as follows: ALT 44 IU/L; ALP 132 IU/L; AST, 38 IU/L; GGT, 106 IU/L; total bilirubin 1.2 mg/dL.

**Table 2 medicina-59-00752-t002:** Baseline characteristics of the cohort according to liver disease etiology.

Variable	Viral Hepatitis	NAFLD/ NASH	ALD	Other	*p*
Age, years (median, range)	49 (21–79)	49 (21–75)	59.5 (45–63)	60 (32–61)	N.S.
Weight, kg (median, range)	79.3 (49–118)	88 (72–125)	81.5 (61–138)	87 (77–102)	0.04
Height, cm (median, range)	172 (156–192)	176 (150–197)	178 (170–186)	172 (165–193)	N.S.
BMI, kg/m^2^ (median, range)	26.0 (18.7–39)	29.0 (21–39)	27.4 (18.4–39.0)	26.9 (23.8–30.1)	0.03
Waist circumference,	90 (66–126)	102 (90–123)	95 (85–144)	91 (90–103)	N.S.
cm (median, range)					
Laboratory data					
ALT, IU/L, median, range	45 (19–356)	60 (18–516)	24 (18–42)	52 (12–81)	0.002
AST, IU/L, median, range	36 (16–236)	36 (15–130)	33 (20–75)	46 (23–53)	N.S.
GGT, IU/L, median, range	51 (9–1130)	49 (17–404)	90 (24–519)	77 (19–107)	0.01
ALP, IU/L, median, range	71 (32–169)	77 (44–212)	75 (60–124)	60 (60–331)	N.S.
Total bilirubin, mg/dl, median, range	0.50 (0.18–1.9)	0.58 (0.18–1.35)	0.50 (0.35–1.69)	0.52 (0.35–0.84)	0.003
Platelets [×10^9^/L], median, range	216 (11–373)	257 (223–267)	300 (66–483)	234 (159–431)	N.S.
Noninvasive serum tests					
APRI score	0.37 (0.13–2.49)	0.55 (0.15–1.44)	0.24 (0.19–3.13)	0.42 (0.22–0.79)	N.S.
FIB-4 score	1.04 (0.2–5.9)	1.10 (0.34–2.75)	1.13 (0.66–10.75)	1.11 (0.42–2.46)	N.S.
ELF score	9 (7.4–11.7)	8.7 (7.3–10.5)	9.4 (9.0–11.5)	9.9 (7.8–12.6)	N.S.
VCTE assessment (Fibroscan^®^)					
LSM	6.7 (4.2–42.2)	6.5 (3.1–54.2)	8.5 (3–44.7)	6.3 (6.1–11.8)	N.S.

ALD, alcoholic liver disease; ALT, alanine aminotransferase; ALP, alkaline phosphatase; AST, aspartate aminotransferase; APRI, AST to Platelet Ratio Index; BMI, body mass index; GGT, gamma-glutamyl transpeptidase; LSM, liver stiffness measurement; NAFLD/NASH, non-alcoholic fatty liver disease/non-alcoholic steatohepatitis; VCTE, vibration-controlled transient elastography. The reference upper limits of normal for liver function tests are as follows: ALT 44 IU/L; ALP 132 IU/L; AST, 38 IU/L; GGT, 106 IU/L; total bilirubin 1.2 mg/dL.

## Data Availability

The datasets used and analyzed in the study are available from the corresponding author by request.

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
