# Peer review of "Transient Elastography as the First-Line Assessment of Liver Fibrosis and Its Correlation with Serum Markers"

_medicina, 2023, doi:10.3390/medicina59040752_

Round 1
Reviewer 1 Report
The authors have assessed the correlation of LSM 15 with serum fibrosis markers to identify patients with significant liver fibrosis. Certain questions required to be addressed are as follows:
· Introduction should include details about the rationale of the study.
· The methodology should include clearly the inclusion and exclusion criteria.
· The discussion should include a paragraph discussing the results of Predictors of significant fibrosis at the first examination in detail as this is important finding.
· Language needs minor corrections.
Author Response
The authors have assessed the correlation of LSM 15 with serum fibrosis markers to identify patients with significant liver fibrosis. Certain questions required to be addressed are as follows:
- Introduction should include details about the rationale of the study.
According to the reviewers’ suggestion, two paragraphs explaining in more detail the rationale of our study were added to the introduction section of the manuscript.
- The methodology should clearly include the inclusion and exclusion criteria.
The inclusion and exclusion criteria for the study were added into the „Materials and Methods“ section.
- The discussion should include a paragraph discussing the results of Predictors of significant fibrosis at the first examination in detail as this is important finding.
A paragraph discussing the predictors and causes of significant fibrosis was added into the Discussion, according to revivewer’s comment.
- Language needs minor corrections.
The text was double-checked for transcription and grammatical errors and revised by a native English speaker.
Reviewer 2 Report
Specific comments to authors: Uzlova and co-others in this research article report on ‘‘Transient elastography as the first-line assessment of liver fibrosis’’. The manuscript is of clinical interest although the authors need to address some points as follows: - Abstract: mention abbreviations in details as first mentioned then use the abbreviation there after. - Assessment of fibrosis using VCTE is validated only for HCV, HBV and AIH and there are different cutoffs of degrees of fibrosis for each one of these diseases not a fixed value as the authors mentioned. - You did not mention any exclusion criteria for your patients that can affect the accuracy of the VCTE as hepatitis or marked hyperbilirubinemia. - What is the range of serum bilirubin in your patients? - Please mention the upper limit of normal of the biochemical parameters used in your study. - Regarding risk factors for significant fibrosis, please add duration of illness, treatment status, presence of co-morbidities. - Does any of your patients have dual diagnosis? e.g. HCV and NAFLD? - It is better to do the correlations between fibrosis scores and results of VCTE in each disease separately. Similarly, mention the result of cutoff age for significant fibrosis in each disease in separate. - Add a table comparing the 4 groups of patients including the demographic data, fibrosis scores, VCTE result and risk factors. - On what basis did you diagnosis patients with NASH? - What does your study add on what is already mentioned in the literature? Thanks
Author Response
Specific comments to authors: Uzlova and co-others in this research article report on ‘‘Transient elastography as the first-line assessment of liver fibrosis’’. The manuscript is of clinical interest although the authors need to address some points as follows:
Abstract: mention abbreviations in details as first mentioned then use the abbreviation thereafter.
All the abbreviation mentioned in the text were explained at their first use and the abbreviation was used consistently thereafter.
- Assessment of fibrosis using VCTE is validated only for HCV, HBV and AIH and there are different cutoffs of degrees of fibrosis for each one of these diseases not a fixed value as the authors mentioned.
Nowadays, we can use VCTE to assess the degree of liver stiffness independently of the aetiology of liver disease. The Fibroscan Interpretation guide gives the cut-off values for the following diagnoses: HBV, HCV, HCV/HIV co-infection, NAFLD, ALD, AIH, and PBC. AIH and PBC patients were not present in our cohort of patients. The cut-off value of 10 kPa is a value in the range of severe fibrosis for all the patients’ diagnoses present in the cohorts. Especially for patients with chronic hepatitis C, 10 kPa represents a value when the patients should remain in HCC screening programmes, even if they are successfully cured and the liver stiffness regresses in further follow-up. The choice of the cut-off value of 10 kPa was described in the text to explain the rationale for this cut-off. Th wording was changed throughout the text and “advanced fibrosis” was used for those who had LSM higher than 10 kPa.
- You did not mention any exclusion criteria for your patients that can affect the accuracy of the VCTE as hepatitis or marked hyperbilirubinemia.
The inclusion and exclusion criteria for the study were added into the „Materials and Methods“ section, according to the reviewer’s suggestion. None of the patients who were enrolled had signs of acute hepatitis and marked hyperbilirubinemia. These pieces of information are now added in the text.
- What is the range of serum bilirubin in your patients?
The serum total bilirubin values were added into Table 1, including median and ranges of the values, all patients had their bilirubin values within the normal ranges.
- Please mention the upper limit of normal of the biochemical parameters used in your study.
The upper limits of normal of the biochemical parameters assessed in the study were added in the caption of Table 1.
- Regarding risk factors for significant fibrosis, please add duration of illness, treatment status, presence of co-morbidities.
All the patients were examined as a part of the first examination of their liver disease, before treatment initiation. Therefore, the duration of their liver disease started at that timepoint and the patients were not treated so far. The comorbidities were nod assessed in detail as far as the VCTE examination was only one of the examinations in the diagnostic procedure, which was led by the referring physician, who concluded the diagnosis.
- Does any of your patients have dual diagnosis? e.g. HCV and NAFLD?
The only dual diagnosis was in patients with HCV/HIV co-infection, as it is mentioned in Table 1.
- It is better to do the correlations between fibrosis scores and results of VCTE in each disease separately. Similarly, mention the result of cut-off age for significant fibrosis in each disease in separate.
- Add a table comparing the 4 groups of patients including the demographic data, fibrosis scores, VCTE result and risk factors.
We are aware of the fact that the patients with different diagnoses should be assesses separately. However, the cohort of our patients is too small to do such an analysis with an appropriate statistical power. Even the largest group of patients with viral hepatitis consisted of 42 patients with HBV, HCV and HIV coinfection, each of these groups could be analysed separately to be the most accurate. The small sample size of our study was described as a limitation in the Discussion section of our manuscript. However, we consider our study as an important insight into liver disease epidemiology in the Czech Republic.
- On what basis did you diagnosis patients with NASH?
The final diagnosis of the liver disease was established by the referring physician, and all the patients in the group NAFLD/NASH were diagnosed per exclusionem, after all liver diseases were excluded.
- What does your study add on what is already mentioned in the literature?
No studies focusing on the use of non-invasive tests in liver fibrosis assessment have been conducted in the Czech Republic so far. Our study not only reflects the number of the patients with significant liver fibrosis at the time of liver disease diagnosis, but also describes the distribution of causes of liver disease at the initial examination in the Czech Republic. This piece of information has been also missing so far for the Czech population.
Round 2
Reviewer 2 Report
Some of the previous comments have not been addressed.
Author Response
Reviewer 2:
Specific comments to authors: Uzlova and co-others in this research article report on ‘‘Transient elastography as the first-line assessment of liver fibrosis’’. The manuscript is of clinical interest although the authors need to address some points as follows:
Abstract: mention abbreviations in detail as first mentioned then use the abbreviation thereafter.
All the abbreviations mentioned in the text were explained at their first use and the abbreviation was used consistently thereafter.
- Assessment of fibrosis using VCTE is validated only for HCV, HBV and AIH and there are different cutoffs of degrees of fibrosis for each one of these diseases not a fixed value as the authors mentioned.
Nowadays, we can use VCTE to assess the degree of liver stiffness independently of the aetiology of liver disease. The Fibroscan Interpretation guide gives the cut-off values for the following diagnoses: HBV, HCV, HC/HIV co-infection, NAFLD, ALD, AIH, and PBC. AIH and PBC patients were not present in our cohort of patients. The cut-off value of 10 kPa is a value in the range of severe fibrosis for all the patients’ diagnoses present in the cohorts. Especially for patients with chronic hepatitis C, 10 kPa represents a value when the patients should remain in HCC screening, even if they are successfully cured and the liver stiffness regresses in further follow-up. The choice of the cut-off value of 10 kPa was described in the text to explain the rationale for this cut-off. The wording was changed throughout the text and “advanced fibrosis” was used for those who had LSM higher than 10 kPa.
- You did not mention any exclusion criteria for your patients that can affect the accuracy of the VCTE as hepatitis or marked hyperbilirubinemia.
The inclusion and exclusion criteria for the study were added into the „Materials and Methods“ section, according to the reviewer’s suggestion. None of the patients who were enrolled had signs of acute hepatitis and marked hyperbilirubinemia. These pieces of information are now in the text.
- What is the range of serum bilirubin in your patients?
The serum total bilirubin values were added into Table 1, including median and ranges of the values.
- Please mention the upper limit of normal of the biochemical parameters used in your study.
The upper limits of normal of the biochemical parameters assessed in the study were added in the caption of Table 1.
- Regarding risk factors for significant fibrosis, please add duration of illness, treatment status, presence of co-morbidities.
All the patients were examined as a part of the first examination of their liver disease, before treatment initiation. Therefore, the duration of their liver disease started at that timepoint and the patients were not treated so far. The comorbidities were not assessed in detail as far as the VCTE examination was only one of the examinations in the diagnostic procedure, which was led by the referring physician.
- Does any of your patients have dual diagnosis? e.g. HCV and NAFLD?
The only dual diagnosis was in patients with HCV/HIV co-infection, as it is mentioned in Table 1.
- It is better to do the correlations between fibrosis scores and results of VCTE in each disease separately. Similarly, mention the result of cut-off age for significant fibrosis in each disease in separate.
- Add a table comparing the 4 groups of patients including the demographic data, fibrosis scores, VCTE result and risk factors.
The cohort was divided, according to the reviewer´s suggestion, into 4 groups. The data are presented in Table 2, which was added to the manuscript. The variables were compared between groups and the correlations of different variables were calculated for each group separately. The paragraph was added to the text: “LSM and NITs correlations according to liver disease aetiology“.
The cut-off age points were calculated for the two largest groups, viral hepatitis AND NAFLD.
The small sample size of our study was described as a limitation in the Discussion section of our manuscript. However, we consider our study as an important insight into liver disease epidemiology in the Czech Republic.
- On what basis did you diagnosis patients with NASH?
The final diagnosis of the liver disease was established by the referring physician, and all the patients in the group NAFLD/NASH were diagnosed per exclusionem, after all liver diseases were excluded.
- What does your study add on what is already mentioned in the literature?
No studies focusing on the use of non-invasive tests in liver fibrosis assessment have been conducted in the Czech Republic so far. Our study not only reflects the number of the patients with significant liver fibrosis at the time of liver disease diagnosis, but also describes the distribution of causes of liver disease at the initial examination in the Czech Republic. This piece of information has been also missing so far.